# C-terminal threonines and serines play distinct roles in the desensitization of rhodopsin, a G protein-coupled receptor

Anthony W Azevedo[1], Thuy Doan[2], Hormoz Moaven[3], Iza Sokal[1], Faiza Baameur[4], Sergey A Vishnivetskiy[4], Kristoff T Homan[5], John JG Tesmer[5], Vsevolod V Gurevich[4], Jeannie Chen[3], Fred Rieke[1,6]*

[1]Department of Physiology and Biophysics, University of Washington, Seattle, United States; [2]Department of Ophthalmology, University of Washington, Seattle, United States; [3]Departments of Cell & Neurobiology and Ophthalmology, Zilkha Neurogenetic Institute, Keck School of Medicine of University of Southern California, Los Angeles, United States; [4]Department of Pharmacology, Vanderbilt University School of Medicine, Nashville, United States; [5]Life Sciences Institute, Departments of Pharmacology and Biological Chemistry, University of Michigan, Ann Arbor, United States; [6]Howard Hughes Medical Institute, University of Washington, Seattle, United States

**Abstract** Rod photoreceptors generate measurable responses to single-photon activation of individual molecules of the G protein-coupled receptor (GPCR), rhodopsin. Timely rhodopsin desensitization depends on phosphorylation and arrestin binding, which quenches G protein activation. Rhodopsin phosphorylation has been measured biochemically at C-terminal serine residues, suggesting that these residues are critical for producing fast, low-noise responses. The role of native threonine residues is unclear. We compared single-photon responses from rhodopsin lacking native serine or threonine phosphorylation sites. Contrary to expectation, serine-only rhodopsin generated prolonged step-like single-photon responses that terminated abruptly and randomly, whereas threonine-only rhodopsin generated responses that were only modestly slower than normal. We show that the step-like responses of serine-only rhodopsin reflect slow and stochastic arrestin binding. Thus, threonine sites play a privileged role in promoting timely arrestin binding and rhodopsin desensitization. Similar coordination of phosphorylation and arrestin binding may more generally permit tight control of the duration of GPCR activity.

*For correspondence: rieke@u.washington.edu

**Competing interests:** The authors declare that no competing interests exist.

## Introduction

G protein-coupled receptors (GPCRs) mediate an array of cellular responses with widely divergent kinetics (*Lefkowitz and Shenoy, 2005*). This diversity originates in part from differences in the processes controlling GPCR activity, which influence the duration and extent to which downstream pathways are activated (*DeWire et al., 2007*). Nonetheless, all but a few GPCRs desensitize through a common set of reactions: phosphorylation by G protein receptor kinases (GRKs) followed by the binding of arrestin. Interestingly, the >800 GPCRs expressed by humans are desensitized by only 7 GRKs and 4 arrestins (*Wilden et al., 1986*; *Lohse et al., 1990*; *Pitcher et al., 1998*; *Gurevich and Gurevich, 2004*). Given this apparent conservation of the desensitization reactions, how is diversity in GPCR lifetime achieved?

GPCR C-termini, where the desensitization reactions take place, typically have multiple possible phosphorylation sites (*Krupnick and Benovic, 1998*). Biochemical experiments indicate that some sites are more rapidly phosphorylated than others, for example, serine sites appear to be preferred in

**eLife digest** 'Rod' cells in the eye enable us to see in starlight. Inside these cells, a protein called rhodopsin is activated by light, which leads to an electrical signal being produced that travels to the brain. The duration of the electrical signal depends on the time it takes for the rhodopsin to be deactivated. Rhodopsin is a member of a large class of receptor proteins known as G protein-coupled receptors that regulate many processes throughout the body.

Previous studies have shown that rhodopsin is deactivated by the attachment of phosphate groups to the protein. This allows another protein called arrestin to bind to rhodopsin. The phosphates can be attached to particular amino acids—the building blocks of proteins—at one end of rhodopsin. Three of these are a type of amino acid called serine. Previous work has shown that light increases the speed at which phosphate groups are added to these serines, suggesting that they are important for producing rapid electrical signals. The other three amino acids are of a different type—called threonine—but it is less clear what role they play in deactivating rhodopsin.

Here, Azevedo et al. studied mutant forms of rhodopsin that were missing the serines or threonines in mice. The experiments show that loss of the serines only slightly slowed the electrical signals. However, loss of the threonines resulted in much slower electrical signals that ended at random times. This was due to rhodopsin being less able to bind to arrestin.

Azevedo et al. propose a new model for how rhodopsin is deactivated. Once light activates the protein, phosphate groups are rapidly added to the serines, which begins to lower the activity of rhodopsin. However, it is the slower addition of phosphates to the threonines that is essential to promote arrestin binding and fully deactivate the protein. Other proteins belonging to the G protein-coupled receptor family also have these serines and threonines, and thus, may be regulated in a similar way.

the case of the photopigment rhodopsin (*Kennedy et al., 2001*; *Maeda, 2003*). The pattern of phosphorylation sites can control the effect of arrestin binding on the receptor, that is, whether arrestin promotes receptor desensitization or internalization (*Key et al., 2003*; *Kim et al., 2005*; *Ren et al., 2005*; *Butcher et al., 2011*; *Nobles et al., 2011*; *Shukla et al., 2011*). The variety of patterns exhibited by different GPCRs has been compared to identification by different bar codes (*Tobin et al., 2008*; *Nobles et al., 2011*). This suggests that, similarly, the pattern of phosphorylation sites may determine the active receptor lifetime, and hence the duration of the resulting signals.

Desensitization of the photopigment rhodopsin provides a unique opportunity to study GPCR desensitization because the signal produced by the activation of a single GPCR can be measured directly (*Baylor et al., 1979*). Furthermore, past work characterizing responses produced by the activation of single rhodopsin molecules has identified an unexpected property of how this GPCR desensitizes: the signals produced by single rhodopsin activations vary much less than expected from other familiar signals controlled by single molecules (*Baylor et al., 1979*; *Rieke and Baylor, 1998*; *Whitlock and Lamb, 1999*; *Field and Rieke, 2002*; *Doan et al., 2006*). This low variability requires precise control of the timing of rhodopsin desensitization (*Mendez et al., 2000*; *Doan et al., 2006*, *2009*). In particular, phosphorylation and arrestin binding need to work in a coordinated fashion so as to minimize brief rhodopsin activations produced by too rapid arrestin binding and long rhodopsin activations produced by too slow arrestin binding. How the required coordination between phosphorylation and arrestin binding works is unclear.

Here, we compare the roles of native serine and threonine sites in rhodopsin desensitization. Our experiments reveal an unexpected specificity of these reactions that underlies the tight coordination of phosphorylation and arrestin binding. Namely, native serine sites are phosphorylated rapidly and begin to lower rhodopsin's catalytic activity (*Ohguro et al., 1993*, *1994*, *1995*; *Gibson et al., 2000*; *Mendez et al., 2000*; *Kennedy et al., 2001*), but serine phosphorylation only weakly promotes arrestin binding; native threonine sites are phosphorylated more slowly but are essential to promote rapid arrestin binding. This coordination produces an orderly sequence of desensitization events and contributes to the precise control of rhodopsin's active lifetime. A similar interplay of serine and threonine sites may be an important general mechanism tuning the kinetics of arrestin binding and GPCR desensitization.

## Results

The results are organized as follows. First, we show that single-photon responses (SPRs) from wild-type (WT), serine-only, and threonine-only rhodopsin differ substantially. Second, we show that, despite the large differences in responses, arrestin binding and receptor phosphorylation remain the critical reactions that desensitize the mutant receptors. Third, we show that the slow step-like responses generated by serine-only rhodopsin can be attributed to slow arrestin binding. Finally, we use the prolonged responses of serine-only rhodopsin to compare the kinetics of downstream steps in the transduction cascade with those of rhodopsin desensitization.

### Serine-only rhodopsin produces prolonged SPRs

*Figure 1A* depicts the canonical molecules and reactions responsible for GPCR desensitization. The native C-terminal sequence of mouse rhodopsin and the mutations examined in this study are also depicted: WT rhodopsin (Rho), rhodopsin with C-terminal serine residues replaced with alanines (S → A), and rhodopsin with C-terminal threonine residues replaced with alanines (T → A). Transgenic mouse strains were backcrossed with Rho$^{-/-}$ mice so that these rods expressed only a single type of rhodopsin (*Lem et al., 1999*).

We recorded responses to brief flashes of light in individual rod outer segments using suction electrodes (*Yau et al., 1977*). The flash strength was adjusted such that rods failed to respond to a majority of the flashes ('failures') but occasionally produced a quantized response. We used quantal analysis to isolate responses produced by the absorption of single photons (*Figure 1B–G*; see 'Materials and methods' and *Figure 1—figure supplement 1* for isolation procedure). Average SPRs elicited by each type of rhodopsin differed, particularly in recovery kinetics (*Figure 1H*, bottom). Time-to-peak and peak amplitude were not significantly different across rhodopsin types (p > 0.14, Kruskal–Wallis test, corrected for multiple comparisons).

Individual responses to single rhodopsin activations (*Figure 1B–G*) reveal several features not apparent in the averages. In particular, T → A rhodopsin (*Figure 1D*) produced responses that plateau and then terminate abruptly. Overlaying all SPRs gathered from a rod illustrates the extent of this effect (*Figure 1G*); whereas responses elicited by Rho or S → A rhodopsin follow similar trajectories, the T → A responses plateau to a common level before terminating at random times. This gives rise to two bands, an upper one that thins over time as responses decay, and a lower one that collects the responses as they terminate. Thus, removal of serine vs threonine sites has quite different effects on SPR variability. Below, we characterize these differences to gain insight into the underlying mechanisms.

### Removing serine and threonine sites increases response variability

We first estimated the duration of each response by determining the time at which the response terminated (*Figure 2A*, insets). Termination times were estimated using an algorithm to identify state transitions, in this case the return of the current to baseline (see 'Materials and methods'). *Figure 2A* (top panel) plots the cumulative distributions of response durations from each recorded cell expressing a given rhodopsin. The pooled distributions are plotted below. Spread in the measurement is dominated by real variation in response duration across strains, as shown by two measures. First, the distributions are wider than the jitter produced by noise in the algorithm (see 'Materials and methods', *Figure 2A*-bottom, gray dots). Second, the distributions of response durations for individual cells of a given strain more closely resemble the pooled distributions for that strain than those of a different strain (*Figure 2A*). The response duration distributions do illustrate an overlap in response properties across strains, for example, the briefest T → A responses are similar to Rho responses both in kinetics (*Figure 1B–D*) and duration (*Figure 2A*). Likewise, Rho responses can occasionally last quite long (*Baylor et al., 1984*).

The time course of response variability was measured from the ensemble variance at each point in time (variance of SPRs minus variance of failures). SPRs followed a consistent initial trajectory and then began to diverge near the time at which the mean response peaked. For all rhodopsin types, the variance peaked after the peak of the mean response (*Figure 2C,D*). This late peak of the variance reflects the fact that the rising phase and peak amplitude of the response vary little compared to the falling phase. The T → A mutant is an extreme case of this effect: the peak in the variance of the T → A responses coincides with the point at which approximately half of the responses have returned to

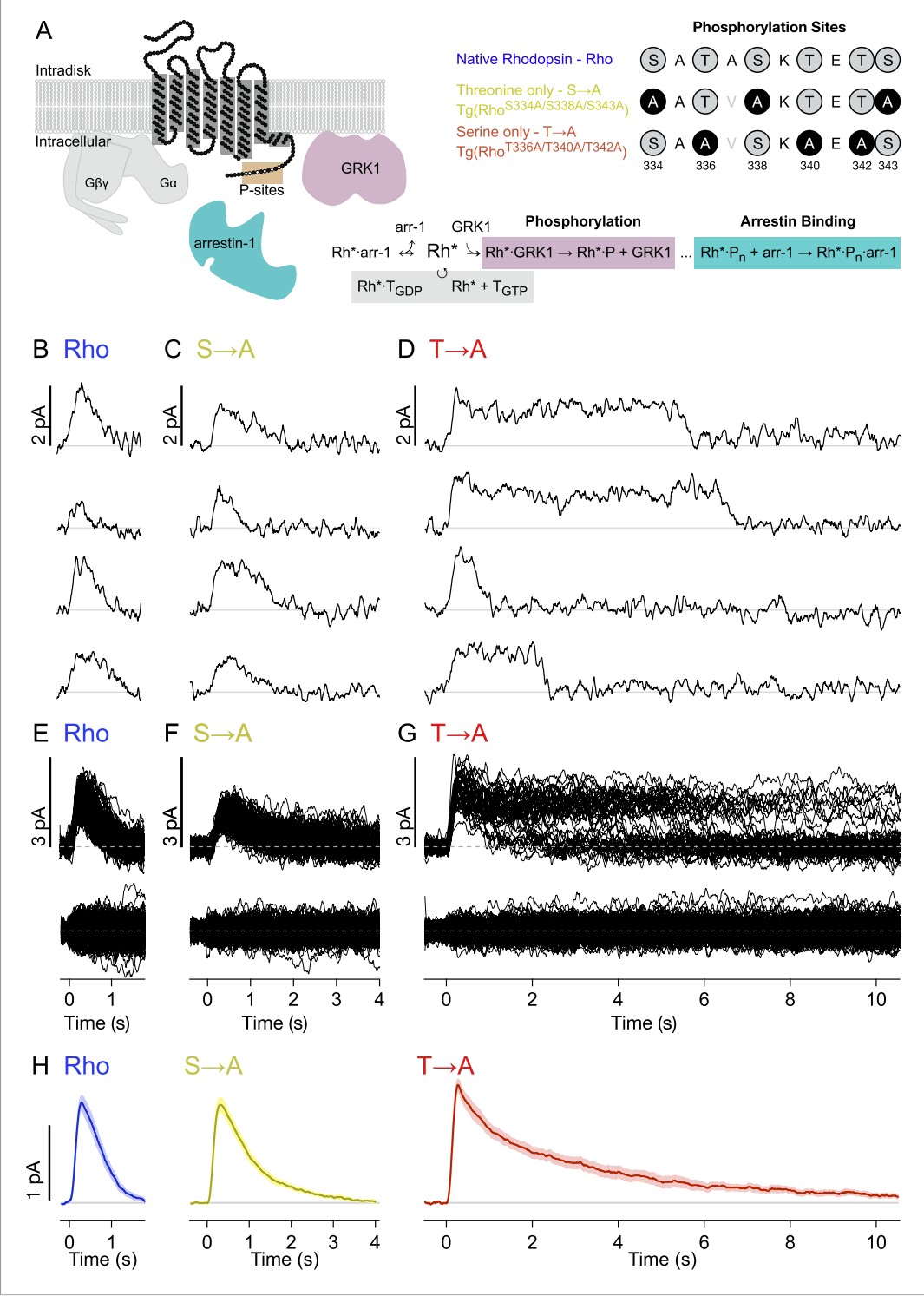

**Figure 1.** Single-photon responses: activation of individual wild-type and mutant rhodopsins. (**A**) Schematic showing rhodopsin, the G protein transducin (T = Gβγ and Gα, gray), G protein receptor kinases 1(GRK1), and arrestin-1. Phosphorylation consists of GRK1 binding and phosphate attachment. Arrestin-1 serves to quench rhodopsin activity and to compete with GRK1, modifying the GRK1 binding rate (*Doan et al., 2009*). The upper right panel shows the sequence of the rhodopsin C-terminus and the mutations to test the effects of replacing either threonine or serine residues with alanine. The mutation A337V is included to produce a linear epitope for mAb 3A6, as in *Mendez et al. (2000)*. Labels indicate strain nomenclature (above) and protein nomenclature (below). (**B–D**) Representative examples of individual single-photon responses (SPRs). (**E–G**) All SPRs from representative cells of each strain, with

*Figure 1. continued on next page*

*Figure 1. Continued*

identified singles (~50) above and failures (no response to flash, ~150) below: (**B**, **E**) Rhodopsin (wild type, WT); (**C**, **F**) S → A, previously referred to as STM (*Mendez et al., 2000*); (**D**, **G**) T → A. (**H**) Average SPRs across all cells: WT—N = 8, S → A—N = 9, T → A—N = 9.
The following figure supplement is available for figure 1:

**Figure supplement 1**. Isolation of single-photon responses.

baseline (2.9 s, compare to median duration 2.7 s in *Figure 2B*). This correspondence between the median response duration and the time at which the variance peaks is expected for a first-order decay process in which response duration is determined by a single stochastic step. Thus, the time course of the variance suggests that a single desensitization step is greatly prolonged for T → A rhodopsin.

Total response variability was quantified using the coefficient of variation (SD/mean) of the integrated SPRs, that is, the coefficient of variation of the response areas ($CV_{area}$). This measure is insensitive to whether rhodopsin desensitization is fast or slow relative to the kinetics of downstream steps in the transduction cascade, whereas the variability of the response at a given time (e.g., the peak) depends on the underlying kinetics (*Hamer et al., 2003*). The variance of the areas of the SPRs was corrected for the variance of the failures and then used to compute $CV_{area}$ (*Hamer et al., 2003*; *Doan et al., 2006*, *2009*). $CV_{area}$ for WT rods (0.37 ± 0.02) was comparable to that found previously (*Doan et al., 2006*). Responses of S → A rods had increased variability ($CV_{area}$ = 0.51 ± 0.02, *Figure 2E*); this increase in $CV_{area}$ is quantitatively consistent with a model in which total WT rhodopsin activity is controlled by a series of ~7 kinetic steps, 3 of which have been removed in S → A rhodopsin (*Doan et al., 2006*). By comparison, the measured increase in variability is greater than that predicted by models that posit that non-linear amplification of the receptor activity contributes substantially to low-response variability (*Caruso et al., 2011*, their table S6).

The $CV_{area}$ for the T → A responses (0.78 ± 0.03, *Figure 2E*) was much higher, consistent with a strong disruption of the normally reproducible timing of rhodopsin desensitization. The different effects of removing threonines and serines were not predicted by previous models of rhodopsin desensitization; instead each kinetic step was assumed to represent identical or at least equivalent biochemical events (*Rieke and Baylor, 1998*; *Hamer et al., 2003*; *Doan et al., 2006*).

Thus, transgenic expression of phosphorylation site mutants produces SPRs with several salient features. S → A rods have longer and more variable single photon-responses than WT rods. T → A rods have much longer and dramatically more variable responses, which exhibit a characteristic plateau followed by an abrupt recovery; variability in the duration of this plateau suggests that a single step dominates the time course of desensitization of T → A rhodopsin. The dependence of the kinetics and variability of the response on both serine and threonine sites indicates that they play distinct roles in controlling rhodopsin desensitization.

## Desensitization of rhodopsin mutants requires phosphorylation and arrestin-1 binding

Do changes in the time course of responses produced by the mutant rhodopsins indicate desensitization through non-canonical mechanisms? As described below, a combination of biochemical and physiological experiments indicates that this is not the case.

Isoelectric focusing uses an immobilized pH gradient in a polyacrylamide gel to separate proteins according to their isoelectric point (pI), which changes with phosphorylation level (*Figure 3A*). We first exposed retinae to a bright 470-nm light to activate all of the rhodopsin (>98%), then froze the retinae to stop all reactions at different time points (0 min, 10 min). Immunoblots of rhodopsin from solubilized WT retina samples showed eight bands, indicating unphosphorylated rhodopsin (0), opsin (0*) plus six differently phosphorylated species, consistent with previous results (*Mendez et al., 2000*). Blots of both T → A and S → A samples showed five bands, corresponding to opsin (0*) and unphosphorylated (0), singly (1), doubly (2), and triply (3) phosphorylated rhodopsin, showing that T → A and S → A mutants can be phosphorylated at all sites (*Figure 3A,B*, *Mendez et al., 2000*). Quantification of the intensity of the bands showed that the T → A mutant had higher levels of doubly and triply phosphorylated rhodopsin than the S → A mutants (*Figure 3B*, *Mendez et al., 2000*), a result

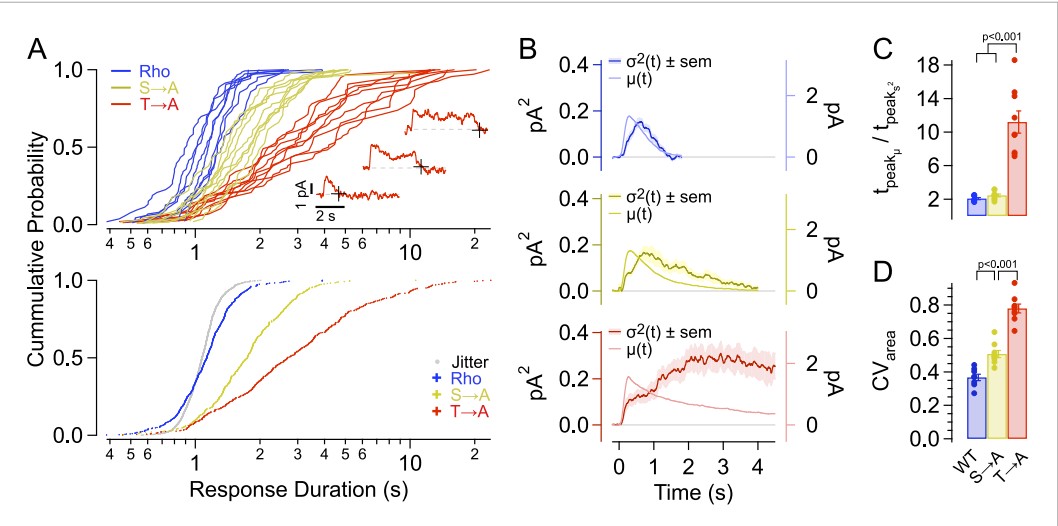

**Figure 2**. Single-photon response properties. (**A**-top) Distribution of response durations for each cell (single line). The response duration was determined by detecting the transition from an 'up' state to a 'down' state, as in *Draber and Schultze (1994)* (Inset). (**A**-bottom) Pooled distributions across cells. Noise in determining response duration shown in gray. Response duration distributions are non-normal ($p < 0.001$, K-S test for non-normality) and are significantly different from one another at the 99% confidence level (Kruskal–Wallis analysis of variance, corrected for multiple comparisons). (**B**) Average time course of SPR variance (left axis) compared to the average SPR (right axis) for each strain. (**C**) Ratio of the time-to-peak of the variance relative to the time-to-peak of the mean response across cells for each strain. WT—$2.1 \pm 0.8$ s; S → A—$2.4 \pm 0.8$ s; T → A—$11.2 \pm 0.8$ s. WT and S → A n.s. (**D**) Coefficients of variation, $CV_{area}$, for individual cells (points) and mean $\pm$ SEM (bars). All measurements are significantly different ($p < 0.001$, one-way ANOVA, corrected for multiple comparisons).

consistent with previous observations that serine sites are more rapidly and efficiently phosphorylated than threonine sites (*Ohguro et al., 1993*, *1994*, *1995*; *Kennedy et al., 2001*).

In addition to the finding that mutant rhodopsins can be phosphorylated, arrestin-1 was able to bind to T → A and S → A rhodopsin in purified outer segment disk membranes that had been phosphorylated by recombinant GRK1 (*Figure 3C*). This finding is also consistent with less direct assays of bovine arrestin-1 binding to rhodopsin mutants expressed and isolated from cultured cells (*Brannock et al., 1999*). As expected from the biochemical assay, the expression of mutant receptors on an arrestin-1 knockout background (Arr$^{-/-}$) prolonged responses (*Figure 3D*); the similarity of these prolonged responses to those of Rho/Arr$^{-/-}$ rods (*Xu et al., 1997*) indicates that termination of the mutant activity requires arrestin-1.

## Response differences are not due to alterations in the transduction cascade

Transgenic protein expression can generate compensatory changes in rod outer segments, such as changes in the concentration of other transduction cascade components or changes in outer segment volume or disk size. Such changes can in turn alter response kinetics (*Calvert et al., 2001*; *Wen et al., 2009*; *Makino et al., 2012*). Several analyses, however, indicate that the differences in SPRs illustrated in *Figures 1, 2* are attributable to the mutations in rhodopsin rather than other factors. We consider first S → A and then T → A rods.

Alterations in rhodopsin expression level affect the ability of a rod to absorb incident light. Such changes can be measured from the 'collecting area', which relates the number of photoactivated rhodopsin molecules produced by a light flash to the photon density of the flash (*Baylor et al., 1979*). The collecting area is proportional to the total amount of rhodopsin, that is, the product of concentration and volume. We measured the collecting area of S → A rods from the probability that a flash of a known photon density failed to produce a response (*Figure 4A*; *Baylor et al., 1979*). The collecting area for S → A rods was $0.63 \pm 0.04$ $\mu m^2$ or 1.3× the collecting area of WT rods ($0.5$ $\mu m^2$, *Azevedo and Rieke, 2011*).

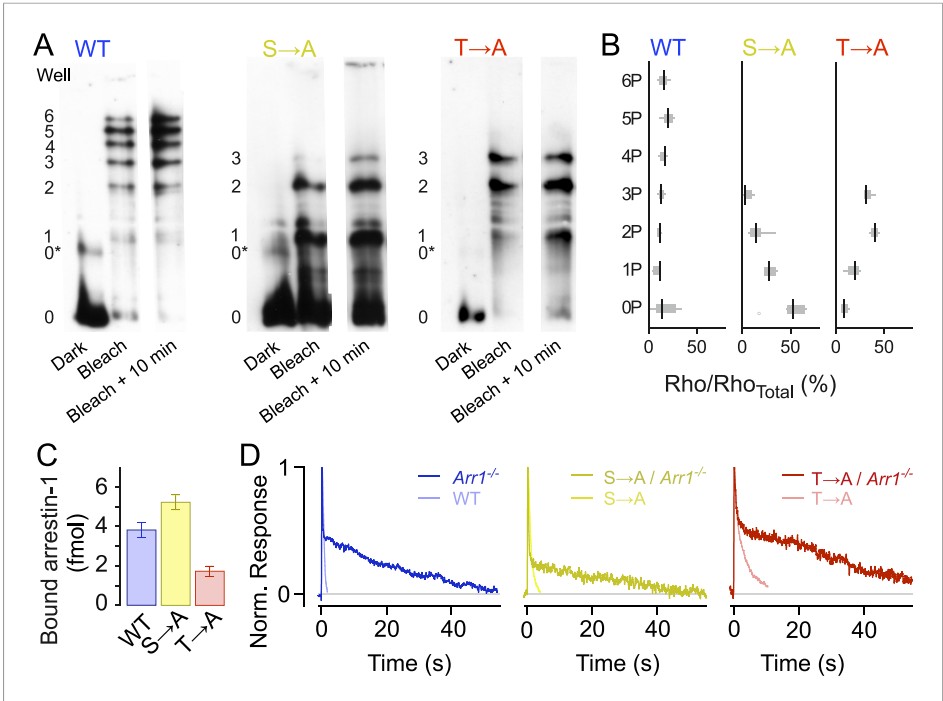

**Figure 3**. Rhodopsin mutants are multiply phosphorylated in vivo. (**A**) Isoelectric focusing of WT, S → A, or T → A retinal membranes from dark-adapted retina (Dark) or retinas exposed to 5-min light (Bleached) or after illumination for 5 min followed by dark incubation for 10 min (Bleached + 10 min). Each strain was run on a separate gel; all lanes in a panel are from the same gel. The gel was blotted and probed with R2-12N, a mouse monoclonal antibody that recognizes the amino-terminus of rhodopsin. The rhodopsin species are indicated at the left of the gel, from unphosphorylated (0) to maximally phosphorylated (6 for WT and 3 for the T → A and S → A mutants). 0* corresponds to unphosphorylated opsin. (**B**) The density of the bands was quantified using the Odyssey Imaging System (LI-COR Biosciences), normalized to total rhodopsin, and plotted as box and whisker plots. Black lines indicate the median density (N = 4–9). (**C**–**D**) Rhodopsin desensitization requires arrestin-1 binding. (**C**) Arrestin-1 binding to rhodopsin in isolated rod outer segment membranes (see 'Materials and methods'). (**D**) Knocking out arrestin-1 prolongs average responses. Each panel compares the average response (8–12 cells in each case) with and without arrestin-1. Average time constants, measured after the initial decay, were 27 ± 2 s for $Arr1^{-/-}$ rods, 20 ± 3 s for S → A/$Arr1^{-/-}$ rods, and 34 ± 4 s for T → A/$Arr1^{-/-}$ rods.

A larger collecting area could be due to an increase in rod volume, rhodopsin concentration, or both. 3D reconstruction of rods from serial electron micrographs showed that transgenic expression of T → A and S → A rhodopsin changed the size of rod outer segments (*Figure 4D*, *Figure 4—figure supplement 1*). The volume of S → A rods was 1.3× larger than WT rods (T → A rods had ∼0.75× the volume). Thus, S → A rods express ∼1.3× as much rhodopsin as WT rods, at the same concentration. A quantitatively similar increase in rod volume has been shown previously to slow the SPR, though not to the extent that S → A SPRs are slowed (*Calvert et al., 2001*; *Wen et al., 2009*; *Makino et al., 2012*). Thus, differences between S → A and WT responses cannot be explained by changes in volume or rhodopsin expression level. As described below (*Figure 8—figure supplement 1*), analysis of the kinetics of response recovery indicates minimal changes in the transduction cascade of S → A rods. Together, these results indicate that differences in S → A and WT responses are due to alterations in the time course of activity of S → A compared to WT rhodopsin.

The differences between Rho and S → A responses here are more subtle than those previously reported (*Mendez et al., 2000*). Rod responses, however, depend substantially on recording conditions (*Azevedo and Rieke, 2011*); indeed, the differences between Rho and S → A were much closer to those observed previously when we replicated the recording conditions used in earlier work (*Figure 4—figure supplement 2*). The rationale for the choice of recording conditions that we used for the bulk of the work here is described in *Azevedo and Rieke (2011)*.

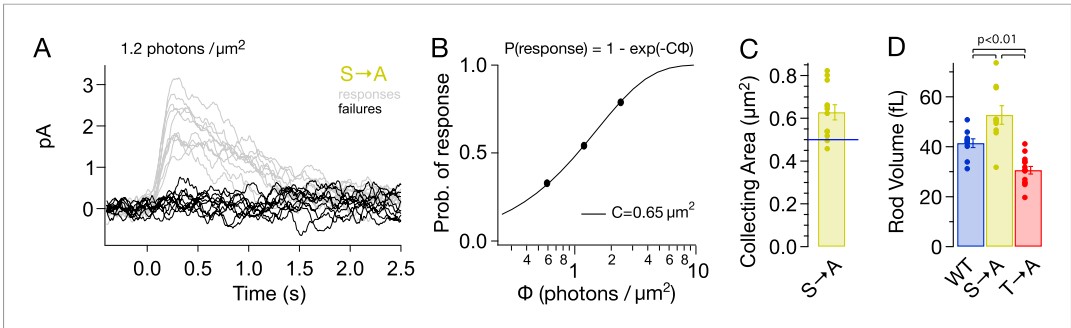

**Figure 4**. S → A rods express ~1.3× total rhodopsin as WT rods, in ~1.3× larger rods, at the same concentration. (**A**) Responses from a S → A rod to a dim, full-field flash, delivering 1.2 photons/μm², classified as either a response (gray) or failure (black). The rod responded 66 times to 121 flashes. (**B**) The fraction of flashes producing responses is plotted against the flash strength for the rod in (**A**). The poisson distribution gives the fraction of responses as p(response) = 1 − p(failure) = 1 − exp(−CΦ), where Φ is the flash strength (photons/μm²), and C is the collecting area (in μm²). A collecting area of 0.65 μm² gives the best fit to the data. (**C**) Across N = 12 cells, the collecting areas measured 0.63 ± 0.04 μm², ~1.3× larger than WT rods (0.5 μm², blue line, *Azevedo and Rieke, 2011*). (**D**) Volume of rod outer segments from 3D serial reconstructions (*Figure 4—figure supplement 1*). WT, 41 ± 2 fL; S → A, 53 ± 4 fL; and T → A, 31 ± 2 fL (p < 0.01, one-way ANOVA, corrected for multiple comparisons).

The following figure supplements are available for figure 4:

**Figure supplement 1**. Ectopic expression of rhodopsin mutants modestly alters rod morphology.

**Figure supplement 2**. Effect of experimental conditions on S → A responses.

To check for compensatory factors in T → A rods, Rho and T → A rhodopsin were expressed in the same rod (T → A/*Rho*[+/−], called 'Combo' rods). Analogous experiments using S → A/*Rho*[+/−] rods were not possible as crossing S → A/*Rho*[−/−] with either WT or T → A/*Rho*[−/−] mice resulted in very few visible rods, likely because of rhodopsin overexpression. Combo rods produced both short- and long-lasting SPRs (*Figure 5A*). Several aspects of these responses could be explained as a mixture of unaltered responses from WT and T → A rhodopsin expressed in roughly equal numbers. (1) The normalized average response of the Combo rods could be fit by a weighted sum of the Rho (51%) and T → A responses (49%) (*Figure 5B*), indicating that a near-equal mixture of the pure responses can explain the time course of the Combo rod responses. (2) The distribution of response durations from the Combo rods (gray crosses in *Figure 5C*) was best fit by an equal mixture of the duration distributions from Rho and T → A rhodopsin (black line in *Figure 5C*). (3) The kinetics of identified T → A-like and Rho-like responses from Combo rods agreed closely with the kinetics of the pure responses (*Figure 5D,E*; see *Figure 5—figure supplement 1* and 'Materials and methods' for response identification).

In summary, *Figure 5* shows that T → A rhodopsin produces responses with similar kinetics in either T → A or Combo rods, and WT rhodopsin produces responses with similar kinetics in either WT or Combo rods. These findings indicate that WT, T → A, and Combo rods have similar transduction cascade kinetics, and thus, that replacement of threonines causes the prolonged SPRs in T → A rods.

## Rhodopsin C-terminal threonine residues are essential for rapid arrestin-1 binding

The experiments described above indicate that the deletion of serine or threonine phosphorylation sites differentially affects rhodopsin desensitization, and that these effects cannot be attributed to either non-canonical desensitization pathways or compensatory changes in the transduction cascade. We next probed the mechanisms responsible for the slowed response kinetics in S → A rods and the abrupt cessation of T → A responses.

We hypothesized that the step-like T → A responses are terminated by the sudden desensitization of the GPCR via either abrupt phosphorylation or arrestin-1 binding. Phosphorylation consists of GRK1 binding and phosphate attachment, both of which can contribute to the rate of desensitization

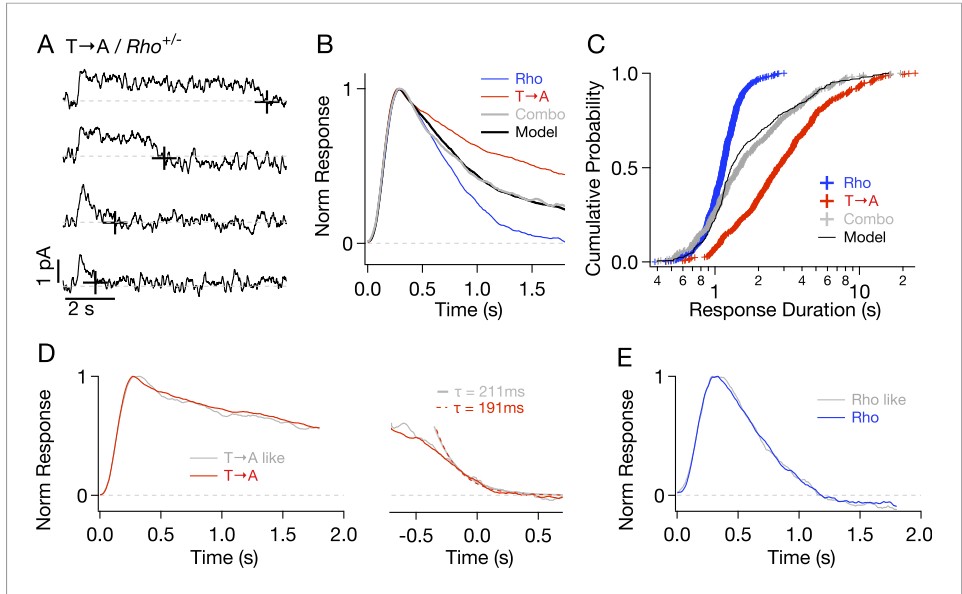

**Figure 5**. Comparison of single-photon responses from different rhodopsin mutants expressed in the same cell. (**A**) Representative SPRs from a T → A/Rho$^{+/-}$ cell, termed a 'Combo' rod. Response termination shown as black crosses. (**B**) Normalized response from Combo rods (gray) fit by a linear combination of normalized Rho and pure T → A responses. (**C**) Cumulative distribution of response durations from pure Rho and T → A rods and from Combo rods. A random selection of Rho and T → A response durations in a 1:1 ratio gives the modeled distribution (black line). (**D**-left) Average of initial phase of T → A responses (red) and combo responses (gray) classified as T → A-like (**Figure 5—figure supplement 1**). (**D**-right) Average tails of T → A-like responses aligned to the response ends, normalized to the peaks (left). Dotted lines depict decaying exponential fits to the combo tail (gray, τ = 211 ms) and T → A tail (red, τ = 191 ms). (**E**) Averages of Rho-like responses from WT and Combo rods.

The following figure supplement is available for figure 5:

**Figure supplement 1**. Combo rod responses were classified according to similarity to a template that discriminated between Rho and T → A responses (inset).

(**Figure 1A**; **Doan et al., 2009**). Arrestin-1 serves both to quench rhodopsin activity and to modify the GRK1-binding rate by competing with GRK1 (**Doan et al., 2009**). The long duration of the T → A responses suggests that GRK1 binding, phosphate attachment, or arrestin-1 binding is substantially slowed and limits the desensitization rate. We distinguished among these possibilities in both T → A and S → A rods by lowering separately the GRK1 or arrestin-1 concentration. If GRK1 binding is rate limiting, lowering the GRK1 but not the arrestin-1 concentration should further slow the responses. If arrestin-1 binding is rate limiting, lowering the arrestin-1 but not the GRK1 concentration should slow the responses. If phosphate attachment by bound GRK1 is rate limiting, the responses should be insensitive to lowering GRK1 and arrestin-1 concentrations.

We genetically lowered the concentrations of arrestin-1 or GRK1 in S → A rods by crossing S → A mice with either *GRK1$^{-/-}$/Rho$^{-/-}$* mice or *Arr1$^{-/-}$/Rho$^{-/-}$* mice to produce heterozygous strains that express less GRK1 or arrestin-1. Western blots confirmed reduced concentrations of arrestin-1 and GRK1 (data not shown). Reduced arrestin-1 expression had little or no effect on S → A SPR kinetics (**Figure 6A,B**, cyan trace) or durations (**Figure 6C**, cyan crosses). Reduced GRK1 expression similarly did not noticeably change SPR kinetics or durations (**Figure 6D–F**). This result differs from reduced GRK1 expression in Rho rods (*GRK1$^{+/-}$/Rho$^{+/+}$*), which clearly exhibit slowed responses (**Doan et al., 2009**; **Chen et al., 2010**). The smaller effect in S → A rods suggests that phosphate attachment by bound GRK1 is much slower than GRK1 binding in these rods—likely a consequence of having only threonine residues remaining (see 'Discussion'). Slow phosphorylation of threonine residues is consistent with previous biochemical findings that serine sites are preferentially phosphorylated (**Ohguro et al., 1993**, **1994**, **1995**; **Kennedy et al., 2001**) and with the relatively low levels of doubly

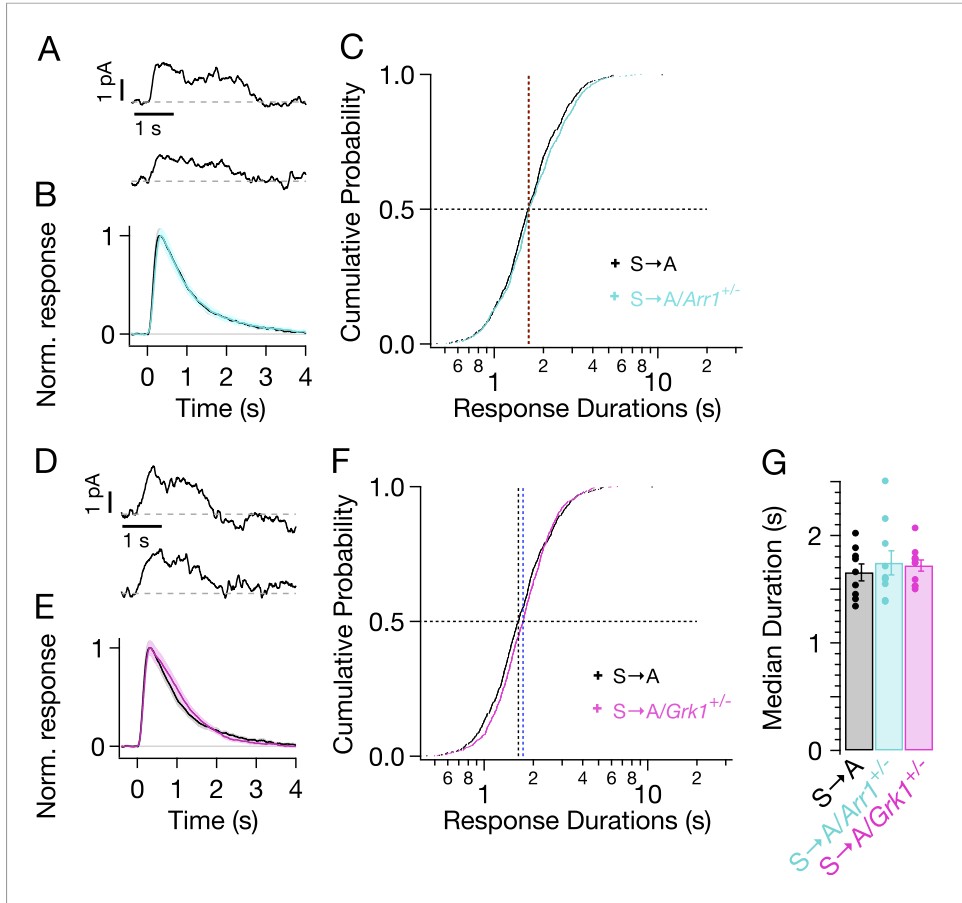

**Figure 6.** Genetic reduction of arrestin-1 or GRK1 expression does not alter S → A responses. (**A**) Representative SPRs in S → A/Arr1+/− rods. (**B**) Average single S → A responses across cells with either normal (black) or lowered (cyan) arrestin-1 concentration. (**C**) Response durations in all cells with either normal (black) or lowered (cyan) arrestin-1 concentration. Pooled distributions are indistinguishable (p > 0.05, Kruskal–Wallis analysis of variance, corrected for multiple comparisons). (**D–F**) Comparison of S → A and S → A/GRK1+/− (magenta) responses and response durations. Pooled distributions are indistinguishable (p > 0.05, Kruskal–Wallis analysis of variance). (**G**) Median response durations with reduced arrestin-1 or GRK1 in individual S → A rods.

and triply phosphorylated rhodopsin in S → A isoelectric-focusing measurements (*Mendez et al., 2000*, *Figure 3A*).

Reduced expression of arrestin-1 in T → A rods led to SPRs that qualitatively resemble those with native arrestin-1 concentrations, that is, a plateau of variable length, followed by a rapid recovery to baseline (*Figure 7A*). The trajectory of the abrupt recovery from the plateau was similar to that of T → A rods, suggesting that the cascade downstream of rhodopsin was unchanged (data not shown). Reducing the arrestin-1 concentration, however, slowed the kinetics of the average SPR (*Figure 7B*, cyan trace). Correspondingly, the duration distribution in T → A/*Arr1*+/− rods shifted to longer times (*Figure 7C*): the median ΔT (50% point on the y-axis) increased by a factor of ~1.5, from 2.7 s to 4.3 s (*Figure 7C,G*). Thus, desensitization of T → A rhodopsin is slowed by reducing the arrestin-1 concentration.

Slow phosphorylation of the remaining serine residues by GRK1 could also contribute to the duration of the T → A responses. Reducing the GRK1 concentration, however, produced modest changes in the average SPR (*Figure 7E*) and response duration (*Figure 7F*). The cumulative probability distribution of T → A/*GRK1*+/− response durations (*Figure 7F*) shows fewer short responses (<1 s), which accounts for the steepening of the curve. This suggests that GRK1 binding is slowed, as predicted by the results of previous experiments with *GRK1*+/− mice (*Doan et al., 2009*). Long responses, and thus the median duration, are little affected (*Figure 7F,G*). The sensitivity of T → A responses to reducing arrestin-1 and

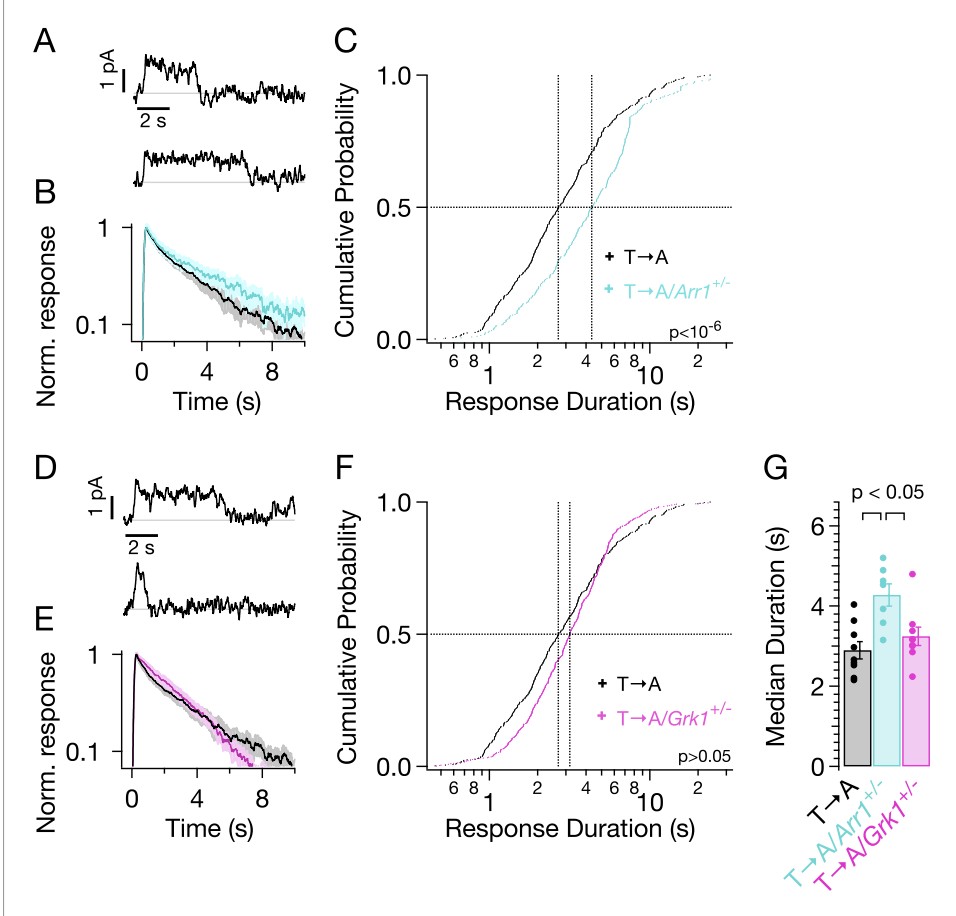

**Figure 7**. Genetic reduction of arrestin-1 expression prolongs T → A responses. (**A**) Representative SPRs in T → A/Arr1+/− rods. (**B**) Average single T → A responses across cells with either normal (black) or lowered (cyan) arrestin-1 concentration, showing slower response decay with lower arrestin-1 (log y-axis). (**C**) Response durations in all cells with either normal (black) or lowered (cyan) arrestin-1 concentration. Pooled distributions are significantly different ($p < 10^{-6}$, Kruskal–Wallis analysis of variance). (**D–F**) Comparison of T → A and T → A/GRK1+/− (magenta) responses. Pooled response duration distributions are indistinguishable ($p > 0.05$, Kruskal–Wallis analysis of variance). (**G**) Median response durations with reduced arrestin-1 or GRK1 in individual T → A rods.

the relative insensitivity to reducing GRK1 implicates arrestin-1 binding as the slow step in desensitization of T → A rhodopsin. This in turn suggests that threonine residues are required for timely rhodopsin desensitization.

## Rhodopsin desensitization and the transduction cascade have similar kinetics

The kinetics of rhodopsin desensitization relative to those of downstream events in the transduction cascade is an important factor shaping SPRs. However, separating the two is difficult (*Pepperberg et al., 1992*; *Rieke and Baylor, 1998*; *Krispel et al., 2006*; *Burns and Pugh, 2010*; *Gross and Burns, 2010*). In one hypothetical regime, rhodopsin activates a collection of G proteins and then quickly desensitizes. In this case, the kinetics of the response recovery reflects the rate at which the active G protein population decays. In another hypothetical regime, rhodopsin desensitizes slowly compared to the rate of G protein decay such that the kinetics of the response recovery reflects the rate at which rhodopsin desensitizes. More generally, the response recovery kinetics reflect both G protein decay and rhodopsin desensitization. Indeed, the analysis described below indicates that

the kinetics of brief SPRs are shaped almost entirely by the transduction cascade, while longer-lasting responses also reflect the kinetics of rhodopsin desensitization.

We grouped WT, T → A, and Combo rod responses according to their measured durations (*Figure 5C*), binning them into 1-s wide bins centered every 0.5 s (e.g., responses with durations <1 s, durations between 0.5 s and 1.5 s, etc). We then aligned the binned responses either to their termination points (*Figure 8A*) or to the time of the flash (*Figure 8C*) and measured the time constants of decay of the average responses in each bin. We plot the termination-aligned decay time constants ($\tau_{term}$; *Figure 8B*) and flash-aligned decay time constants ($\tau_{flash}$; *Figure 8D*) as functions of the response duration for WT, T → A, and Combo responses. The Combo responses, as might be expected from *Figure 5*, are consistent with a combination of WT and T → A responses.

This analysis highlights several response properties. First, the fastest response decay has a time constant ~200 ms, similar to previous measurements of the rate of the G protein's GTPase activity (*Krispel et al., 2006*). Three types of responses share this fast decay: (1) all termination-aligned responses (*Figure 8B*); (2) the briefest flash-aligned responses (*Figure 8C*; 0.5 s bin); and (3) the longest flash-aligned responses (*Figure 8C*; 4 s bin). Flash-aligned traces of intermediate kinetics have substantially slower decay time constants (*Figure 8D*, $\tau_{flash}$; bins between 1 s and 3 s). This effect was consistent for responses of Rho and T → A rhodopsin, including when both were in the same cell.

Termination-aligned S → A responses also exhibited recovery time constants near 200 ms (*Figure 8—figure supplement 1*), matching those of termination-aligned responses from Rho and T → A rhodopsin. The similarity of these time constant supports our conclusion that S → A transduction is normal.

The analysis of *Figure 8* suggests that G protein decay dominates the kinetics of the fastest WT responses, when, stochastically, rhodopsin is short-lived. The slower recovery of flash-aligned, intermediate-duration WT responses is consistent with a combination of slower rhodopsin desensitization and G protein decay. This effect is clear in the T → A responses (*Figure 1G*), where the slow decay of the average response reflects the long and variable response plateau; the analysis here indicates a similar picture that applies to WT responses.

To noticeably impact the recovery kinetics, rhodopsin must desensitize with comparable or slower kinetics than G protein decay. Consistent with this suggestion, the dashed lines in *Figure 8B,D* plot the decay time constant when all WT SPRs are averaged, the typical measure of the recovery time constant. The time constant of this average response is considerably slower than the fastest recovery time constants of the binned responses. Thus, our data support a model in which stochastic rhodopsin desensitization occasionally occurs quickly, such that the transduction cascade kinetics dominate the response kinetics, but desensitization often occurs more slowly and influences response kinetics.

## Discussion

The importance of phosphorylation and arrestin binding for GPCR desensitization has been appreciated for several decades (*Stadel et al., 1982*, *1983*; *Wilden and Kühn, 1982*; *Kühn et al., 1984*; *Maeda, 2003*). Nonetheless, we lack a clear understanding of how these mechanisms support the observed diversity in GPCR signaling. Here, we investigated the role of different C-terminus phosphorylation sites in rhodopsin desensitization. We found that phosphorylation of native serines is rapid but ineffective in promoting rapid arrestin-1 binding. Phosphorylation of native threonines is slow but effective in promoting rapid arrestin-1 binding. Together, these properties ensure not only that several phosphorylation steps will complete before arrestin-1 will bind but also that after these steps complete, arrestin-1 will bind rapidly. These results challenge assumptions about the relative importance of serine and threonine sites and show how phosphorylation and arrestin binding can act in concert to tightly control the timing of rhodopsin desensitization.

### Asymmetric roles of serine and threonine residues in rapid GPCR desensitization

An important concept for understanding desensitization reactions has been that accumulation of local charge at the C-terminus triggers arrestin activation and binding (*Gurevich and Gurevich, 2004*). Interestingly, normal arrestin-1 binding was not recapitulated in vitro by substituting charged residues for native phosphorylation sites (*Ling et al., 2004*; *Ascano and Robinson, 2006*; *Ascano et al., 2006*). Our results show that the precise residues at which charge accumulates have distinct mechanistic

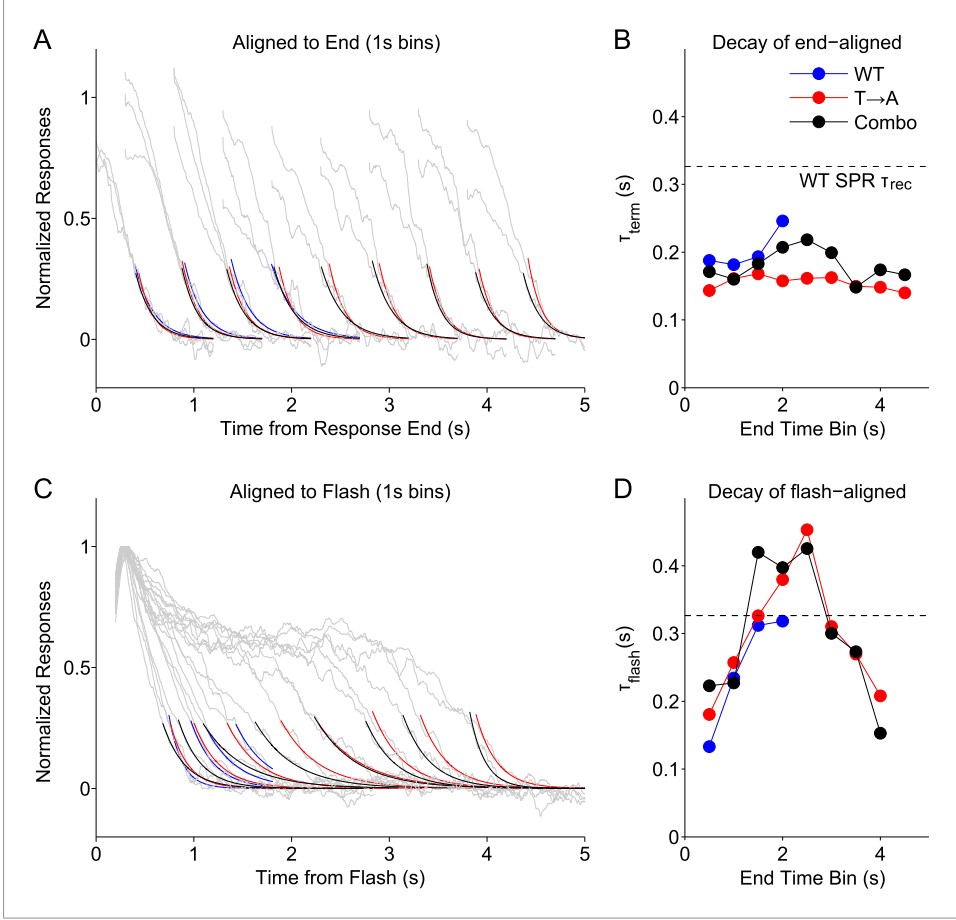

Figure 8. Stochastic rhodopsin desensitization affects the kinectics of single-photon responses. (A) Responses from WT, T → A, and Combo lines were binned according to their duration, in 1-s wide bins every 0.5 s (oversampling) and aligned to their termination points (gray lines). An exponential function (A*exp(−t/B)) was fit to the decline (<0.25) (blue, WT; red, T → A; black, combo). (B) Time constant of termination-aligned decay ($\tau_{term}$) as a function of the bin center. Dashed line shows the recovery time constant of the average WT SPR. (C) Responses were aligned to the time of the flash and fit with the decay function. (D) Time constant of decay for the average of responses in each 1-s bin, aligned to the flash.
The following figure supplement is available for figure 8:

Figure supplement 1. Termination-aligned S → A responses exhibit recovery time constants of ~200 ms.

impacts on the desensitization process—further supporting the idea that specific residues and their phosphorylation are required.

Removal of C-terminal threonine sites (T → A) compromised arrestin-1 binding and dramatically prolonged SPRs. However, rhodopsin with only three serine sites remaining (T → A) recovered with a mean duration of ~4 s (Figure 2A), more than five times faster than responses from rhodopsin with only two serine sites remaining (S334/S338) (Doan et al., 2006). Thus, the addition of a single serine (at position 343) substantially speeds the response, likely because of faster arrestin-1 binding (Figure 7; Vishnivetskiy et al., 2007). This suggests that serine sites promote arrestin-1 binding, albeit much less effectively than threonine sites.

Removal of native C-terminal serine residues (S → A) modestly slowed SPRs and increased their variability. The kinetics of responses generated by threonine-only rhodopsin were insensitive to reducing the concentration of either GRK1 or arrestin-1. This behavior differs from the clear sensitivity of WT SPRs to reductions in either GRK1 or arrestin-1 (Doan et al., 2009). Among the reactions

controlling rhodopsin desensitization (*Figure 1A*), only the rate of phosphate attachment is insensitive to GRK1 and arrestin-1 concentration. Hence, the rate of desensitization of threonine-only rhodopsin appears to be limited by phosphate attachment by bound GRK1 rather than GRK1 or arrestin-1 binding. This is consistent with biochemical experiments indicating that threonine sites are more slowly phosphorylated than serine sites (*Ohguro et al., 1993*, *1994*, *1995*; *Kennedy et al., 2001*).

## Comparison with biochemical studies of phosphorylation

Biochemical experiments implicate serines as preferred phosphorylation sites for rhodopsin (*Ohguro et al., 1993*, *1994*, *1995*; *Kennedy et al., 2001*). This work has led to a model in which rhodopsin is phosphorylated sequentially, beginning with the distal most serine (S343), where the C-terminus is most mobile and likely to encounter GRK1 (*Langen et al., 1999*) and proceeding toward the most proximal site (*Kennedy et al., 2001*). More recent experiments have detected threonine phosphorylation (*Lee et al., 2002*, *2010*; *Vishnivetskiy et al., 2007*), particularly in the absence of serine residues (*Mendez et al., 2000*). The isoelectric-focusing experiments described here (*Figure 3A*) are consistent with a preference for serine over threonine phosphorylation.

Rapidity of phosphorylation, however, need not equate with rapid arrestin binding. Several additional lines of evidence show that threonine phosphorylation preferentially promotes arrestin-1 binding. Maximizing the affinity of arrestin-1 binding has been found to require four or more phosphates, suggesting that phosphorylation of at least one threonine is necessary for normal desensitization (*Vishnivetskiy et al., 2007*). It also appears that regardless of where threonines are located in the C-terminal sequence, their presence leads to comparable degrees of receptor quenching in a cell-free assay (*Brannock et al., 1999*). Other GPCRs can function in a similar manner: the somatostatin subtype 2A receptor is phosphorylated at serine sites to a greater extent than at threonine sites (*Hipkin et al., 1997*), but threonine phosphorylation is required for normal desensitization (*Liu et al., 2008*). Thus, the asymmetric role of threonine and serine sites may be a general mechanism tuning the affinity and binding kinetics of arrestin, controlling timely and reproducible desensitization of GPCR activity, and determining which intracellular pathways are activated.

Recent studies have advanced our understanding of the interactions between GPCRs and arrestin at the protein structure level. Our findings suggest that threonines play a privileged role in these interactions. Arrestins are composed of a C-domain and an N-domain, connected by a 12-residue 'hinge' region. These two domains are normally held fixed by the interaction of the arrestin C-terminus with the N-domain and the polar core, a network of charges paradoxically buried within the protein (*Granzin et al., 1998*; *Hirsch et al., 1999*). The phosphorylated C-terminus of the GPCR is suspected to disrupt the polar core, freeing the C-terminus and gate loops (*Hirsch et al., 1999*; *Gurevich and Gurevich, 2004*). Once free, arrestin's N-domain rotates relative to the C-domain, changing the relative positions of loops that form the receptor-binding interface, and perhaps permitting more dynamic movements (*Kim et al., 2013*; *Shukla et al., 2013*, *2014*; *Zhuang et al., 2013*; *Vishnivetskiy et al., 2013a*; *Gurevich and Gurevich, 2014*). In the context of this model, our results suggest that rhodopsin's native threonine sites more readily interact with arrestin's polar core to 'activate' arrestin. Other recent evidence points to arrestin's '139 loop' as a critical mediator of selectivity for phosphorylated rhodopsin (*Vishnivetskiy et al., 2013a*). A crystal structure of arrestin-2 (β-arrestin-1) together with a phosphorylated C-terminus peptide from the V2R receptor shows that a critical lysine in the 'middle loop' (139 loop) is coordinated with a phosphate (*Shukla et al., 2013*). In this model, our data indicate that the arrangement of rhodopsin's C-terminal phosphorylation sites is important for this interaction.

## Reproducibility of the SPR

Most signals controlled by single molecules exhibit large trial-to-trial variability. Familiar examples are the charge flowing through an ion channel during a single opening and the time-to-decay of a radioactive particle. In these examples, variability results from stochastic fluctuations in the active lifetime of the molecule responsible. SPRs generated by vertebrate rods vary much less than expected from simple stochastic models for single molecules (*Baylor et al., 1979*; *Rieke and Baylor, 1998*; *Whitlock and Lamb, 1999*; *Field and Rieke, 2002*; *Doan et al., 2006*, *2009*; *Gross et al., 2012*). Thus, the effective lifetime of rhodopsin's catalytic activity appears to be tightly regulated.

While tight regulation is required to explain low variability in rhodopsin's lifetime, this does not constrain the average kinetics of rhodopsin desensitization. Rhodopsin's effective lifetime in mouse rods has been a matter of debate: some studies support fast rhodopsin desensitization, such that G protein decay determines the rate of response recovery (*Krispel et al., 2006*; *Gross and Burns, 2010*). In this case, variability in the timing of rhodopsin desensitization produces variable amplitude SPRs, with less variability in response time course (*Gross et al., 2012*). Other studies find that the rate of rhodopsin desensitization is sufficiently slow to affect the rate of response recovery (*Doan et al., 2009*). In this case, the time course of the SPRs will also vary. The different observations across studies appear to be due in part to differences in experimental conditions and the physiological state of the rods (*Azevedo and Rieke, 2011*). The results of *Figure 8* indicate that, under our experimental conditions, rhodopsin activity is occasionally short-lived but often persists sufficiently long to shape the SPRs (see also *Baylor et al., 1984*). This suggests that on average the kinetics of rhodopsin desensitization are closely matched to the kinetics of the transduction cascade.

In principle, low variability, or reproducibility, of the SPR can be achieved by having multiple steps in rhodopsin desensitization; averaging across kinetic steps reduces variability in rhodopsin's cumulative activity, and hence, in the SPR (*Rieke and Baylor, 1998*). Assuming that the transduction cascade acts linearly to convert rhodopsin activity to changes in current, replicating the measured low variability of the response requires 6–7 steps (*Doan et al., 2006*). Non-linearities in the transduction process could reduce the number of steps required (*Bisegna et al., 2008*; *Gross et al., 2012*); nonetheless, a shared feature of all current explanations for reproducibility is that multiple events must work in a concerted fashion to control rhodopsin desensitization. In particular, arrestin-1 binding must be coordinated with phosphorylation so that it does not occur too quickly or too slowly.

Our results here provide a clearer biophysical picture for how such coordination could occur: serine sites are phosphorylated rapidly and begin to reduce rhodopsin's catalytic activity, but phosphorylation of serines is ineffective in promoting rapid arrestin-1 binding; threonine sites are phosphorylated more slowly, but promote rapid arrestin-1 binding. This model predicts few short-lived rhodopsin activations, since arrestin-1 would rarely bind until most of the C-terminus sites had been phosphorylated. Similarly, the model predicts few long-lived active molecules since arrestin-1 binds rapidly once threonine sites are phosphorylated. Thus, the different roles of C-terminus serine and threonine phosphorylation sites appear to ensure that the first events in vision produce low-noise quantal signals.

## Materials and methods

### Mouse transgenesis and breeding

All animal procedures were approved by the Administrative Panel on Laboratory Animal Care at the University of Washington. Wild-type mice (C57B/6J, called here WT for convenience) were purchased from Jackson Laboratory. Transgenic mice were created as previously described (*Mendez et al., 2000*). Briefly, the 11-kB BamHI fragment containing rhodopsin gene and the 6-kb upstream were cloned into the pBII SK(−) plasmid (all restriction enzymes, New England Biolabs, Ipswich, MA). The BspEI-PacI fragment including the coding region of the C-terminus was excised and mutagenized by PCR (Qiagen) to encode alanines in place of either all three serines (Tg(Rho$^{S334A/S338A/S343A}$) or S → A) or all three threonines (Tg(Rho$^{T336A/T340A/T342A}$) or T → A). In addition, the mutagenized PCR product encoded both the mutation A337V to produce a linear epitope for mAb 3A6, which recognizes the bovine rhodopsin C-teminus and silent EagI sites for genotyping by Southern blot. The A337V mutation is present in transgenic mice with single phosphorylation-site substitutions; these rods have more normal response kinetics, indicating that the A337V substitution does not contribute strongly to the changes described in this paper (*Mendez et al., 2000*; *Doan et al., 2006*). The resulting product was sequenced before ligation with the plasmid and verified later. The construct was linearized by digestion with BamHI and purified for oocyte injection (Qiagen). Injection and implantation was carried out by the University of Washington Transgenic Resource Program into F2 hybrid zygotes from C57Bl/6J and DBA/2J F1 mice and yielded 8 viable S → A founders and 10 viable T → A founders. The first generations were genotyped for the presence of the transgene, and those strains determined to have successfully passed on the transgene were crossed with the rhodopsin knockout strain (Rho$^{−/−}$) for two generations (*Lem et al., 1999*). Primers were: Tg(Rho/T → A) forward: 5′-CGGCCGCCGTTTCCAAGGCGGAGG-3′,

reverse: 5′-GGAGCCTGCATGACCTCATCCC-3′ (157 bp) and Tg(Rho/S → A) forward: 5′-TCCACTGG-GAGATGACGACGCGGC-3′, reverse: 5′-TGAGGGAGCCTGCATGACCTCATCC-3′ (180 bp).

Ultimately, we sought offspring with two alleles with neomycin cassettes (indicating $Rho^{-/-}$) and one wild-type sequence in the same region, indicating the presence of the transgene. Quantitative PCR (Transnetyx) determined the relative amounts of *neo* and *rho* sequence and pups with the proper 2:1 ratio were again crossed with the $Rho^{-/-}$ strain. Primers for PCR were $Rho^+$ and $Rho^-$ forward: 5′-GTGCCTGGAGTTGCGCTGTGGG-3′, $Rho^+$ reverse: 5′-GGCAAAGAAGCCCTCGAGATTACAGCC-3′ (450 bp); $Rho^-$ reverse: 5′-CGGTGGATGTGGAATGTGTGCGAG-3′ (250 bp).

Finally, we dissected retinas to find strains in which rods had not degenerated due to over- or under-expression of transgenic rhodopsin, ultimately yielding a single S → A strain and three T → A strains. The T → A and S → A designations imply $Rho^{-/-}$, that is, transgenic rhodopsin without endogenous rhodopsin.

Nomenclature for the visual arrestin-1 gene, according to Jackson Lab Mouse Genome Informatics, is Sag. We felt that for clarity it would be appropriate to use the gene synonym, Arr1. Therefore, $Sag^{-/-}$ mice are called $Arr1^{-/-}$ mice in this study. Similarly, for the protein SAG, we adopt arrestin-1, the standard term in the GPCR literature. $Arr1^{-/-}/Rho^{-/-}$ and $Grk1^{-/-}/Rho^{-/-}$ double-knockout strains were crossed with S → A or T → A mice to create strains with lowered arrestin-1 expression levels. All transgenic mice used in the experiments were dark-reared to prevent rod degeneration. Responses of dark- and light-reared C57B/6J mice were similar and have been combined.

## Suction electrode recording

We recorded from rods using the suction electrode technique as previously described (*Azevedo and Rieke, 2011*). Mice were dark adapted for >12 hr before they were killed, and retinas were dissected under infrared illumination. Retinas were stored in Ames' medium, equilibrated with 95% $O_2$/5% $CO_2$, at 32℃. For recording, a fraction of a retina was shredded and allowed to adhere to the bottom of the recording chamber before perfusing with Ames' medium at 30℃. Rod outer segments were drawn under negative pressure into a fire-polished microelectrode with a tip bore of 1.4 μm. Light stimuli were generated by blue LEDs (peak at 475 nm) and delivered to the preparation through the microscope condensor. Stimuli consisted of dim 10-ms flashes that alternated between 1× and 1.4× of a nominal flash intensity that was judged to rarely elicit responses, thus ensuring that most responses were to a single rhodopsin isomerization. The dim flashes were spatially restricted by an adjustable slit ~1-μm wide, set perpendicular to the length of the rod outer segment, illuminating ~30–40 disks (*Field and Rieke, 2002*).

Rods were selected for extended recording if they were not obviously damaged by the suction procedure and if their dark currents exceeded 8 pA. Response kinetics and dark currents were checked periodically, and cells were rejected if either had changed. Acquisition software written in Igor (Wavemetrics) controlled stimuli and collected data. Basic measurements of response parameters are reported in *Table 1*.

Each trial, also called an epoch, consisted of a pre-period, a 10-ms stimulus, and a response period, followed by a 91-ms pause while data were written to memory. The long stochastic responses seen in T → A mice required striking a balance between response periods long enough to capture most responses in one trial and gathering enough trials. Therefore, response periods were chosen to be long enough to capture ~80% of the responses within one trial (5 s for T → A, 7.5 s for T → A/$Arr1^{+/-}$ rods). Responses that stretched across trials were dealt with as described below.

**Table 1**. Population measurements (±SEM) of dark currents and single-photon response (SPR) parameters for pure strains

|  | Dark current (pA) | SPR peak (pA) | SPR peak time (s) | SPR $\tau_{rec}$ (s) | SPR mean duration (s) | SPR $CV_{area}$ |
|---|---|---|---|---|---|---|
| WT | 10.3 ± 0.2 | 1.35 ± 0.09 | 0.30 ± 0.01 | 0.32 ± 0.04 | 1.19 ± 0.06 | 0.37 ± 0.02 |
| S → A | 8.8 ± 0.5 | 1.34 ± 0.09 | 0.32 ± 0.02 | 0.90 ± 0.06 | 1.84 ± 0.08 | 0.51 ± 0.02 |
| T → A | 10.7 ± 0.7 | 1.55 ± 0.07 | 0.27 ± 0.01 | 3.61 ± 0.44 | 3.96 ± 0.28 | 0.78 ± 0.03 |

$\tau_{rec}$: recovery time constant.

Previous work found larger differences between the average dim-flash responses of S → A and WT rods (*Mendez et al., 2000*). Several factors could contribute to these different findings. First, we regenerated the S → A strain, beginning with the purification of the original constructs, thus, potentially altering rhodopsin expression, rod morphology, and response kinetics. Second, the electrical recordings were carried out using different tissue storage and culture media, factors that have been shown to have a large effect on SPR amplitude and kinetics (*Azevedo and Rieke, 2011*). Indeed, when we replicated a subset of experiments using conditions similar to those used previously, we found larger differences in S → A and Rho responses (*Figure 4—figure supplement 2*).

## Data analysis

Data were analyzed in MATLAB (Mathworks). Isolation of singles involved three steps: determining whether, on a given trial, the flash elicited zero, one or more rhodopsin isomerizations; determining when thermal events occurred; and linking responses that occasionally stretched across epochs.

As previously described (*Field and Rieke, 2002*; *Doan et al., 2006*, *2009*), each trial was assigned a whole number value (0, 1, or 2) that indicated whether the flash produced a 'failure', a 'single', or the response to multiple rhodopsin isomerizations. In brief, classification of responses was based on the correlation of each trial with the average response. These correlations across flashes were fit with a model that identified the most likely number of rhodopsin isomerizations for each individual response. All cells retained for analysis had a clear distinction between failures and singles, with a probability of misclassifying a failure as a single of <1% (*Figure 1—figure supplement 1B*).

Thermal activation of rhodopsin produces current responses identical to those produced by the absorption of a photon, though almost never at the time of the flash. We removed these trials to the extent possible by filtering the measure current records with the average response and looking for events that were within one standard deviation of the amplitude of the SPR.

Software limitations prevented continuous recording of long events in the T → A rods. Additional errors arose when a noise event towards the end of one trial continued to the next and interfered with the response in that trial. Thus, we linked trials containing responses that spanned the inter-trial interval and eliminated trials that did not start at baseline due to previous noise events.

One measure used to quantify responses was the distribution of response durations. We estimated the response duration using an algorithm developed to measure transition times of single-channel currents (*Draber and Schultze, 1994*). Briefly, histograms of current values across SPRs in a given cell typically exhibited two peaks, one centered around 0 pA and another at ~1 pA where the responses plateau, which approximate a 'down state' and an 'up state', respectively. Transitions from the up to down state were identified based on a transition parameter that depended on the baseline noise. Noise could cause the algorithm to detect multiple transitions, so we required the downstate to persist for at least 300 ms. Transitions were typically detected at a point before the response returned to baseline, so the response was deemed to end when the current dropped below the average of points in a 100-ms window following the identified transition time.

The accuracy of the response duration estimate is impacted by the extent to which baseline noise corrupts the estimate of the downward transition. This effect can be measured by adding the average SPR to the time course of trials identified as failures, and measuring the durations, which will vary only as a result of noise (shown as dotted gray line in *Figure 2A*-bottom). Fifty failures from each WT rod make up this distribution in an attempt to represent the full range of baseline noise effects.

The duration algorithm detected the point when the response dropped below a threshold, which usually occurred when the noise caused a downward swing. Aligning the responses to the end points caused noise in multiple traces to become correlated, confounding our estimate of the recovery time constant. To avoid this effect, we averaged noise trials aligned to concocted end points, identified as above, which suffered the same correlation effect. The noise trace was then subtracted from the aligned singles to eliminate the coherent-noise effect and produce a smooth average falling phase (*Figure 8*). This procedure did not inherently produce fast time constants as in *Figure 8B*, because when the average SPR was added to noise trials, the falling phase of the end-aligned simulated responses had a time constant of 307 ms, similar to that of the average SPR (data not shown). Singles with lifetimes within 200 ms of the end of their epochs were excluded from this analysis.

## Isoelectric focusing

Isoelectric focusing was carried out as previously described (*Mendez et al., 2000*) with slight modifications. Retinas were dissected as above, the retinal pigment epithelium removed from the entire retina, and the retina was flat-mounted on a poly-lysine-coated coverslip (BD). The tissue was then illuminated by a calibrated spot of light from an LED with a peak wavelength of 475 nm (Digikey). A 5-min stimulus was estimated to bleach >98% of the visual pigment under these conditions. Following the stimulus, the retina was returned to Ames' medium for an incubation period before being placed in an Eppendorf tube and frozen in ethanol and dry ice. Gel casting, tissue homogenization and solubilization, electrophoresis and immunostaining were as previously described.

## Electron microscopy and reconstruction

To prepare retinas for electron microscopy, we anesthetized mice at 5 weeks of age with sodium pentabarbitol for perfusion fixation through the heart with a solution of 2.5% glutaraldehyde, 2% formaldehyde, 0.08 M $CaCl_2$ in 0.1 M cacodylate buffer. Following fixation, eyes were removed and kept overnight in fixative at 4°C. Half eyes were washed $5 \times 5$ min in Na cacodylate, placed in osmium tetroxide 1%, buffered to 7.2 in Na cacodylate, for 1.5 hr on ice, followed by washing $5 \times 5$ min at room temperature in water. The tissue was placed in 1% urinal acetate in water for 20 min and then washed in water for $4 \times 5$ min. Blocks were then sequentially dehydrated in 50, 70, 90, 95% ethanol, and finally in 100% for $2 \times 15$ min. Dehydration proceeded with $2 \times 15$ min propalene oxide, then with 1:1 propalene oxide, epon araldite for 2 hr, and then 1 hr epon araldite. The resin was changed and allowed to sit for 1 hr. Blocks were then polymerized overnight at 60°C.

Images of serial sections taken near the optic nerve were obtained on a JEOL 1230 scope or a Phillips CM10 scope at 2950× magnification and scanned into a computer. The images were then analyzed using TrakEM2 software (*Cardona et al., 2012*). The default parameters for TrakEM2's non-linear elastic alignment algorithm were sufficient to align the images, producing some deformations that are likely to affect our measurements only slightly and similarly across preparations. Individual rods were manually segmented, and the resulting area lists measured according to TrakEM2 measuring routines.

## Preparation of urea washed ROS

Mice were maintained in darkness overnight before retina dissection, and all procedures were carried out under infrared illumination. Retinal tissue from 6 animals were placed in 150 µl of 8% OptiPrep in Ringer's buffer (10 mM Hepes, pH 7.4, 130 NaCl, 3.6 mM KCl, 2.4 mM $MgCl_2$, 1.2 mM $CaCl_2$, and 0.02 Mm Ethylenediaminetetraacetic acid). Samples were vortexed and centrifuged at 200×*g* for 1 min. The supernatant was removed and stored on ice. The whole procedure was repeated five times. Combined supernatants were layered on a 10–30% continues gradient of Optiprep in Ringer's buffer. The gradient was centrifuged for 50 min at 26,000×*g*. Collected rod outer segments (second band from the top) were diluted three times in Ringer's buffer and centrifuged for 30 min at 30,000×*g*. The pellet was then homogenized in 50 mM Hepes, pH 8.0 containing 5 M urea and centrifuged at 150,000×*g* for 45 min. Membranes were washed three times with Ringer's buffer, and pellet containing urea washed ROS was frozen at −80°C for further use.

## Phosphorylation and arrestin-1 binding

Bovine GRK1 was purified from baculovirus-infected High-5 cells as previously described (*Singh et al., 2008*). The protein sequence was truncated at residue 535, followed by the sequence VDHHHHH to allow for purification by $Ni^{2+}$-NTA chromotography. Phosphorylation reactions were carried out in kinase buffer (20 mM Tris-HCl, pH 7.5, 2 mM $MgCl_2$, 1 mM Dithiothreitol or DTT, 0.5 mM ATP), with rhodopsin (0.6 µM), 11-cis retinal (3.75 µM), and GRK1 (1.6 µM) for 30 min at 22°C under constant illumination (*Vishnivetskiy et al., 2013b*). In vitro transcription and translation of radio-labeled murine arrestin-1 was carried out as described (*Gurevich and Benovic, 1993*). For the arrestin-binding assay, radio-labeled arrestin-1 (2 nM) was incubated with 0.3 µg of rhodopsin in 50-µl buffer (50 mM Tris-HCl, pH 7.4, 100 mM potassium acetate, 1 mM EDTA, 1 mM DTT) for 5 min at 30°C, under room illumination (*Vishnivetskiy et al., 2007*). The samples were cooled and bound arrestin-1 was separated from free arrestin-1 on Sephadex G-100 columns at 4°C. Arrestin-1 binding was quantified by liquid scintillation counting.

## Western blot and protein expression quantification

Retinal proteins from WT mice and heterozygous knock-out mice were separated on 4–15% gradient sodium dodecyl sulfate polyacrylamide gel and transferred onto Immobilon-P membranes. Membranes after blocking with 5% milk were probed with anti-arrestin and anti-β-tubulin polyclonal antibodies (NE Biolabs) with subsequent treatment with alkaline phosphatase-linked anti-rabbit polyclonal antibodies. Signals from developed Western blots were evaluated by scanning densitometry using ImageJ software, and results were determined as background-corrected absorbance units.

## Acknowledgements

We thank the UW Transgenic Core and Dr Bryce Sopher for help generating the transgenic mice, and Dr Jane Sullivan for reagents and assistance; Richard Ahlquist for managing the mouse colony; Dan Possin, Jing Huang, and Ed Parker of the UW Vision Imaging Module (EY01730) for technical assistance; and Dr Rachel Wong for guidance in imaging photoreceptors. Paul Newman, Daniel Carleton, Mark Cafaro assisted with acquisition and analysis software. We also thank members of the Rieke lab for helpful discussions and to Drs Charles Asbury, Bertil Hille, James Jeanne, and John Tuthill for comments on the manuscript. Support provided by HHMI (FR), NIH (EY11850 to FR, T32GM-07270 (AWA), T32EY-07031(TD), EY11500 to VVG, EY12155 to JC, HL071818, and HL086865 to JJGT), a Poncin Scholarship (TD), and American Heart Association post-doctoral fellowship N014938 to KTH.

## Additional information

### Funding

| Funder | Grant reference | Author |
|---|---|---|
| Howard Hughes Medical Institute (HHMI) | | Fred Rieke |
| National Institutes of Health (NIH) | EY11850 | Fred Rieke |
| National Institutes of Health (NIH) | T32GM-07270 | Anthony W Azevedo |
| National Institutes of Health (NIH) | T32EY-07031 | Thuy Doan |
| National Institutes of Health (NIH) | EY11500 | Vsevolod V Gurevich |
| National Institutes of Health (NIH) | EY12155 | Jeannie Chen |
| National Institutes of Health (NIH) | HL071818 | John JG Tesmer |
| National Institutes of Health (NIH) | HL086865 | John JG Tesmer |
| Poncin Scholarship | | Thuy Doan |
| American Heart Association (AHA) | post-doctoral fellowship N014938 | Kristoff T Homan |

The funders had no role in study design, data collection and interpretation, or the decision to submit the work for publication.

### Author contributions

AWA, TD, VVG, JC, Conception and design, Acquisition of data, Analysis and interpretation of data, Drafting or revising the article; HM, FB, SAV, Conception and design, Acquisition of data, Analysis and interpretation of data; IS, Isolated rhodopsin outer segments, Acquisition of data; KTH, JJGT, Expressed and purified GRK1, Acquisition of data; FR, Conception and design, Analysis and interpretation of data, Drafting or revising the article

### Author ORCIDs

Vsevolod V Gurevich, http://orcid.org/0000-0002-3950-5351

## Ethics

Animal experimentation: This work was performed in strict accordance with the recommendations in the Guide for the Care and Use of Laboratory Animals of the National Institutes of Health. All procedures followed protocols approved by the Institutional Animal Care and Use Committee (protocol 3030-01) of the University of Washington.

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
