## [Decision Letter]

Thank you for sending your work entitled “C-terminal Threonines and Serines Play Distinct Roles in the Desensitization of Rhodopsin, a G protein-Coupled Receptor” for consideration at *eLife*. Your article has been favorably evaluated by a Senior editor and three reviewers, one of whom, Ronald L Calabrese, is a member of our Board of Reviewing Editors.

The Reviewing editor and the other reviewers discussed their comments before we reached this decision, and the Reviewing editor has assembled the following comments to help you prepare a revised submission.

The authors present a thorough electrophysiological and biochemical analysis of the kinetics of rhodopsin desensitization in genetically altered mice in which C terminal serines and threonines of rhodopsin are removed (S > A or T > A). The primary observation is that serine-only Rhodopsin gives rise to long duration highly variable quantal responses, whereas the kinetics of quantal responses for threonine-only Rhodopsin shows only modest alteration. The results further indicate that phosphorylation of serines is rapid compared to threonines but that arrestin binding is rapid only after thereonine phosphorylation.

The authors develop a model that posits that serine sites are phosphorylated rapidly, but these events are ineffective in promoting rapid arrestin binding; threonine sites are phosphorylated more slowly, but promote rapid arrestin binding. This model predicts tight control of the duration of quantal light responses thus ensuring that the first events in vision produce low-noise quantal signals.

The paper is well written and the logical flow is very clear. The data set is extensive and well-illustrated. The findings should be of wide general interest as they impact our understanding of desensitization of all GPCRs.

Consensus concerns:

1) The paper is highly technical in many aspects and contains much in the way of supplemental data. Important aspects of the supplemental data should be incorporated into the text for clarity and flow (e.g. Figure 2–figure supplement 4; Figure 3–figure supplement 1; Figure 4—figure supplement 2).

2) There is real concern about expression levels in the mutants used. The authors remade StoA mutant strain, probably as a control to their TtoA mutant mice. Because Rhodopsin level determines response sensitivity, kinetics, and rods' health, it is disappointing that expression level and homogeneity of expression are not characterized. The StoA mutant probably express too many Rhodopsins as the “combo” breeding scheme resulted in “few visible rods”, indicative of unhealthy rods undergoing substantial degeneration at time of analysis. This is really the biggest caveat with the work and interpretation, because the data is interpreted that threonine phosphorylation is slow but triggers quick arrestin binding, while serine phosphorylation is fast but inefficient in triggering arrestin binding, a claim that must be supported by sufficient biochemical characterization of StoA mutant mice.

3) There is an on-going debate in the field regarding whether transducin's active lifetime or rhodopsin's active lifetime is the dominant time constant in the light-response's decay, with the senior author in the present paper leaning toward the latter. In this paper, the opposite viewpoint/finding by the other side of the debate (35) is quote/utilized, but exactly how the current manuscript helps in resolving this puzzle is not clearly spelled out. The sentence “Replicating the measure low variability […] suggests that many of the steps are provided by phosphorylation (16)” (in the subsection “Reproducibility of the single photon response”) is simply too short to deal with this point (see also above). This important issue should be elaborated more.

Germane to this debate is a specific comment by one of the reviewers: “Under physiological condition(s), (the) rod single photon response is stereotypic because the recovery is governed not by the deactivation of one single R* molecule but by the average lifetime of many activated transducin, controlled by the level of GAP complex. I don't understand why the authors ignore this very simple concept. In this MS, R* deactivation is of course a deciding factor because they deliberately slowed it down by making StoA and TtoA mutations, so now variability of R* deactivation, confounded in this work by degeneration issue and transgenic expression level, shows up in response duration and kinetics.”

---

## [Author Response]

*1) The paper is highly technical in many aspects and contains much in the way of supplemental data. Important aspects of the supplemental data should be incorporated into the text for clarity and flow (e.g. Figure 2–figure supplement 4; Figure 3–figure supplement 1;*
Figure 4—figure supplement 2*)*.

Thank you. We have rearranged the contents of the figures to shift information from supplemental to primary figures, particularly in cases where that involved little additional conceptual overhead. This has substantially reduced the number of figure supplements. We also rearranged some of the material in the primary figures to improve flow. Specific changes to the figures were:

a) Moved average responses from Figure 2 to Figure 1;

b) Simplified Figure 1—figure supplement 1 to focus on the selection criteria for cells to include in the analysis;

c) Removed Figure 2–figure supplements 1 and 2 and incorporated key panels into other figures (new Figure 4 and Figure 4—figure supplement 2);

d) Removed Figure 2–figure supplement 3;

e) Added panel A of Figure 2–figure supplement 4 to Figure 2 itself and removed figure supplement;

f) Added Figure 3–figure supplement 1 to Figure 3 and removed figure supplement;

g) Moved key point from Figure 4—figure supplement 2 to Figure 5 and removed the figure supplement.

We made corresponding changes to the text. Those include moving technical details from previous figure supplements into the methods (e.g. the state-transition method used to measure of response durations). We also now describe all of the remaining figure supplements in the text at appropriate locations.

These changes reduced the total number of figure supplements from 9 to 5 and simplified several of those remaining. We also think these changes streamlined the text and substantially reduced the amount a reader needs to jump between figures to follow the logic in the paper.

2) There is real concern about expression levels in the mutants used. The authors remade StoA mutant strain, probably as a control to their TtoA mutant mice. Because Rhodopsin level determines response sensitivity, kinetics, and rods' health, it is disappointing that expression level and homogeneity of expression are not characterized. The StoA mutant probably express too many Rhodopsins as the “combo” breeding scheme resulted in “few visible rods”, indicative of unhealthy rods undergoing substantial degeneration at time of analysis. This is really the biggest caveat with the work and interpretation, because the data is interpreted that threonine phosphorylation is slow but triggers quick arrestin binding, while serine phosphorylation is fast but inefficient in triggering arrestin binding, a claim that must be supported by sufficient biochemical characterization of StoA mutant mice.

This is indeed an important concern and one that we did not adequately deal with in the original manuscript. We have now added and/or highlighted three controls indicating that neither rhodopsin expression level nor possible compensatory changes in transduction cascade components are key factors in the results we report.

First, we have measured the collecting areas of recorded StoA rods, which provide an accurate measure of total rhodopsin content; these experiments are reported in a new Figure 4. We prefer this measure to biochemical assays because it is made specifically on those rods that we select for physiological recordings rather than providing an average across all rods. Collecting areas are ∼1.3x larger in StoA compared to WT rods (Figure 4). The increase in collecting area can be accounted for by the larger volume of StoA rods, such that StoA rhodopsin is expressed at similar concentrations as Rho in WT rods (Figure 4). The change in outer segment volume is too small to account for differences between the light responses of StoA and TtoA rods and between WT and StoA rods, as now noted in the text (please see the subsection entitled “Response differences are not due to alterations in the transduction cascade”).

Second, we compared StoA and WT responses using both our typical recording conditions and those used for recordings from the original StoA strain (the original strain was no longer available). The differences we see from our new strain, when using comparable recording conditions, are similar to those seen in [48]. This is documented in Figure 4—figure supplement 2 and in the the subsection “Response differences are not due to alterations in the transduction cascade”. We also refer to our previous work (3) that explains in detail the reason we chose the recording conditions used in the paper.

Third, we have analyzed the recovery trajectories of the StoA responses using the same analysis previously included for TtoA and WT rods (see Figure 8—figure supplement 1). This analysis shows that StoA responses, when aligned to their recovery times, terminate with near-identical trajctories as TtoA and WT responses. This similarity indicates that the underlying kinetics of the transduction cascade is very similar across all strains studied here. This similarity argues against compensatory changes in StoA rods that could explain their slower light responses.

*3) There is an on-going debate in the field regarding whether transducin's active lifetime or rhodopsin's active lifetime is the dominant time constant in the light-response's decay, with the senior author in the present paper leaning toward the latter. In this paper, the opposite viewpoint/finding by the other side of the debate (*[35]*) is quote/utilized, but exactly how the current manuscript helps in resolving this puzzle is not clearly spelled out. The sentence “Replicating the measure low variability […] suggests that many of the steps are provided by phosphorylation (*[16]*)” (in the subsection “Reproducibility of the single photon response”) is simply too short to deal with this point (see also above). This important issue should be elaborated more*.

Thank you. We have expanded the description of the experiments describing the recovery kinetics and their implications for rhodopsin’s lifetime both in the Results (“Rhodopsin desensitization and the transduction cascade have similar kinetics”) and in the Discussion (“Reproducibility of the single-photon response”). In brief, the trajectory of the final response recovery depends on the response duration, with the slowest final recovery trajectories occurring at intermediate response durations and the fastest final recoveries for the shortest and longest response durations. This dependence suggests that the final recovery trajectory is dominated by recovery of components of the transduction cascade downstream of rhodopsin (most likely transducin) for the shortest and longest duration responses, but that rhodopsin desensitization contributes to the recovery trajectory for intermediate duration responses. We hope this analysis and interpretation is now clear.

*Germane to this debate is a specific comment by one of the reviewers: “Under physiological condition(s), (the) rod single photon response is stereotypic because the recovery is governed not by the deactivation of one single R* molecule but by the average lifetime of many activated transducin, controlled by the level of GAP complex. I don't understand why the authors ignore this very simple concept. In this MS, R* deactivation is of course a deciding factor because they deliberately slowed it down by making StoA and TtoA mutations, so now variability of R* deactivation, confounded in this work by degeneration issue and transgenic expression level, shows up in response duration and kinetics*.*”*

Although it is true that many transducins are activated by a single rhodopsin molecule, this does not explain reproducibility in any of the lines examined (including WT). The number of transducins activated should reflect rhodopsin’s active lifetime. Thus, trials in which rhodopsin is short-lived should produce fewer transducin activations than trials in which rhodopsin is long-lived. If the duration of rhodopsin’s activity is brief compared to the response, then variability in the number of transducins activated will produce variability in the response amplitude; the response time course, on the other hand, will vary little since it will be determined by transducin decay and hence reflect averaging across activated transducins. If the duration of rhodopsin’s activity is comparable to the response, then the resulting dispersion in times at which transducins are activated, together with variability in the number activated, will cause the response to vary in amplitude and in shape.

These issues are the reason why we use the response areas to quantify reproducibility: it captures both of these cases. Indeed, assuming the transduction cascade responds linearly and noiselessly to rhodopsin activity, the CV of the response areas is equal to the CV of the cumulative rhodopsin activity, independent of the relative lifetimes of rhodopsin and transducin. Variability associated with the finite number of transducins activated will only increase response variability and hence cause the CV of the response areas to underestimate reproducibility of rhodopsin itself. We have expanded our discussion of these issues in both the Results (“Removing serine and threonine sites increases response variability”) and Discussion (“Reproducibility of the single-photon response”).